# Unpacking the Complexity of COVID-19 Fatalities: Adverse Events as Contributing Factors—A Single-Center, Retrospective Analysis of the First Two Years of the Pandemic

**DOI:** 10.3390/v15071430

**Published:** 2023-06-24

**Authors:** Aleksander Zińczuk, Marta Rorat, Krzysztof Simon, Tomasz Jurek

**Affiliations:** 1Department of Forensic Medicine, Wroclaw Medical University, 50-369 Wroclaw, Poland; marta.rorat@umw.edu.pl (M.R.); tomasz.jurek@umw.edu.pl (T.J.); 2Department of Infectious Diseases and Hepatology, Wroclaw Medical University, 50-369 Wroclaw, Poland; krzysztof.simon@umw.edu.pl

**Keywords:** SARS-CoV-2, death, mortality, treatment, ARDS

## Abstract

In a retrospective analysis of 477 fatal COVID-19 cases hospitalised at a single medical centre during the period from 6 March 2020 to 30 June 2022, several factors defining those patients at admission were assessed, as well as the course of the hospitalisation and factors contributing to death. There was a predominance of men (59.3% (283)) burdened by comorbidities, with increased inflammation at admission. Patients aged ≥ 81 years were significantly more likely to be admitted to and die in infectious diseases units (IDU) due to respiratory failure, their hospital stays were shorter, and they were most likely not to receive specialist treatment. The most common COVID-19 complications included acute kidney injury in 31.2% (149) patients and thromboembolic complications in 23.5% (112). The course of hospitalisation was complicated by healthcare-associated infections (HAI) in 33.3% (159) of cases, more often in those treated with baricitinib (*p* < 0.001). The initial use of an antibiotic, although common (94.8% (452)), was unwarranted in almost half of the cases (47.6% (215)). Complications of hospitalisation (46.1% (220)) and adverse events involving staff (49.7% (237)) were found in almost half of the patients. In 88.7% (423) of the cases, death was due to respiratory failure in the course of SARS-CoV-2 infection. Adverse events during hospitalisation should be considered as an additional factor that, in addition to the infection itself, may have influenced the death of patients.

## 1. Introduction

SARS-CoV-2 infection has become a global public health threat within a short time of the appearance of the first cases. The pandemic, which has lasted more than three years, resulted in more than 676 million confirmed cases of COVID-19, including more than 6.8 million deaths from the disease—data from 2 May 2023 [1]. The sudden emergence of a large number of patients with severe disease caused by the new coronavirus variant, which was associated with its high infectiousness and ability to cause severe respiratory failure and other complications, led to the overloading of healthcare systems and the occurrence of many excess deaths worldwide as also caused by reduced access to healthcare for patients with other medical conditions [2]. The severity of SARS-CoV-2 infection and the risk of death in its course depends on many factors, the most important of which are considered to be male sex, age > 65 years, the presence of comorbidities, primarily with chronic cardiovascular, respiratory and renal diseases, malignancies, obesity, diabetes and autoimmune conditions, and the variant of the virus causing COVID-19 disease [3,4,5,6]. The World Health Organization (WHO) has been monitoring circulating SARS-CoV-2 variants since January 2020, and so far, five VOCs have been identified—variants of concern (Alpha, Beta, Gamma, Delta and Omicron)—that are responsible for the pandemic waves [7]. The severity of the infection and mortality varies according to the VOCs that cause the infection, with particularly significant differences found between the Delta and Omicron variants [8,9].

The onset of the COVID-19 pandemic was particularly difficult for healthcare personnel, and the risk of adverse events during the provision of healthcare services, including those resulting in death, was particularly high [10]. The rapidly spreading infection quickly exposed the inadequacies of healthcare systems worldwide, including shortages of staff, equipment and funding. An urgent search began for the therapeutic and preventive measures necessary to stop the avalanche of infections. Initially, in a haphazard manner and without adequate scientific research, the treatment used included existing antiviral therapies, antibiotics and drugs used for rheumatic conditions (e.g., azithromycin, lopinavir, ritonavir, oseltamivir, amantadine and chloroquine). Such interventions not only failed to provide benefits but proved to be detrimental to patients in some cases [11]. Subsequently, the efficacy of the following drugs and substances were evaluated: convalescent plasma; remdesivir, a drug designed to treat EBOV (Ebola virus) infection; and drugs with immunomodulatory effects (e.g., dexamethasone, tocilizumab) [12,13]. As the pandemic continued, subsequent recommendations from scientific societies were based on reports containing increasingly better scientific data. The next step in the fight against infection was the introduction of vaccines and new therapies targeted directly against SARS-CoV-2 (monlupiravir, nirmatrelvir/ritonavir) [14,15,16]. Knowledge of the course of COVID-19, the mechanisms of action of the virus and the complications associated with it has evolved rapidly due to the very high number of new publications; from the beginning of the pandemic until 1 April 2023, more than 350,000 publications appeared in the PubMed database under the keyword “COVID-19”. Epidemiological data from the entire pandemic period show that severe infection can affect approximately 20% of patients, and mortality does not exceed 2% [1,17]. Patients with a severe course of the disease are most often hospitalised and are the first at risk of dying from COVID-19, encouraging the conduct of research in this group to reduce mortality. On the other hand, a detailed assessment of the course of hospitalisation makes it possible to detect factors that are not directly related to the virus itself but significantly increase the risk of death.

This retrospective study’s main objective is to analyse fatal COVID-19 cases to determine the risk factors and causes of death, including adverse events.

## 2. Materials and Methods

The retrospective analysis included the medical data of 477 fatal COVID-19 cases out of a total of 2671 patients hospitalised from 6 March 2020 to 30 June 2022 in the infectious diseases units (IDU) and intensive care unit (ICU) of the J. Gromkowski Specialist Regional Hospital in Wrocław (Poland). From the beginning of the pandemic, this hospital was dedicated to patients with suspected and confirmed SARS-CoV-2 infection. In all patients, infection was initially confirmed by a real-time molecular RT-PCR (reverse transcription–polymerase chain reaction) test. Then, due to the acquisition of new validated diagnostic methods, the infection was confirmed by a molecular test or antigen test (PanBio COVID-19 Ag Rapid Test Device by Abbott). Patients admitted from or transferred to other hospitals were excluded from the analysis when their full medical records were not available. A division of the pandemic wave into three periods was used: pre-Delta (1 March 2020–30 June 2021), Delta (1 July–31 December 2021) and Omicron (1 January–30 June 2022), which was based on the dominant SARS-CoV-2 virus sequences in Poland as analysed by GISAID (Global Initiative on Sharing All Influenza Data) [8,18].

The baseline assessment of patients included the following demographic data: age, sex and comorbidities (cardiovascular diseases, respiratory diseases, diabetes, obesity, kidney diseases, malignancies). During SARS-CoV-2 infection, the following data were assessed: duration of symptoms before admission (days) and capillary saturation [19]. An evaluation of the following laboratory parameters were included in the baseline characteristics: C-reactive protein (CRP), procalcitonin (PCT), D-dimer, creatinine and ferritin levels, white blood count (WBC), platelet count (PLT), lactate dehydrogenase (LDH) activity and endothelial activation and stress index (EASIX). The formula proposed by Luft et al. (LDH [U/L] × creatinine [mg/dL]/platelet count [G/L]) was used for the calculation of EASIX [20].

The course of hospitalisation was extensively analysed as follows: length of hospitalisation, oxygen therapy used (oxygen therapy, high-flow nasal oxygenation therapy (HFNOT), non-invasive ventilation (NIV), invasive mechanical ventilation (IMV)); treatment administered, such as antivirals (RDV, remdesivir; MPV, molnupiravir), immunomodulatory (dexamethasone, tocilizumab, baricitinib), convalescent plasma, the type of combination therapy used at the beginning of the pandemic (chloroquine and LPV/r—lopinavir boosted with ritonavir) and the use of low-molecular-weight heparin (LMWH) (enoxaparin or nadroparin) in a prophylactic or therapeutic dose. In the next step, events that occurred during hospitalisation were assessed: coinfections, such as community-acquired infections and healthcare-associated infections (HAIs), which were divided into bloodstream infections (BSIs), urinary tract infections (UTIs) and ventilator-associated pneumonia (VAP). HAIs were diagnosed as defined—an infection that occurred during the provision of medical services and was not present during hospital admission [21]. Questionable culture results that bore the hallmarks of sample contamination (numerous cultured bacteria, organisms that are a component of human microbiota and are detected in only one sample) were excluded from the analysis. The appropriateness (clinical signs, laboratory parameters, microbiological results) of the antibiotic therapy used was analysed in detail. Adverse events were assessed using criteria in line with the current WHO Patient Safety Report, where an adverse event is an event that results in avoidable harm to the patient [22]. The adverse event included both COVID-19 complications (the occurrence of thrombotic complications, such as venous thromboembolism, ischaemic stroke, myocardial infarction and limb ischaemia; bleeding and other complications, such as acute kidney injury (AKI), exacerbation of chronic heart disease and decompensation of liver function) and complications unrelated to the infection but resulting from hospitalisation (HAIs, emphysema and other ventilator-associated complications, trauma or falls; personnel malpractice, such as a lack of or delay in specialist treatment as assessed on an individual basis in accordance with the national or global recommendations for the treatment of COVID-19 patients) [23,24,25]. Finally, the causes of death of the patients were assessed by distinguishing between the main cause of death and other concomitant causes of death.

The study was positively approved by the Bioethics Committee of the Wroclaw Medical University (Decision no. KB-826/2020 dated 17 December 2020). The retrospective nature of anonymised medical data meant that informed consent from patients was not required. 

## 3. Statistical Analysis

Descriptive statistics were presented using means, standard deviations, medians and quartiles for quantitative variables and counts with percentages for qualitative variables. The normality of the data was assessed using the Shapiro–Wilk test. Due to the non-normality of the data, non-parametric tests were used. The relationship between qualitative and quantitative variables was assessed using the Mann–Whitney U test or the Kruskal–Wallis test for two or three groups, respectively. The qualitative analysis was performed using Fisher’s exact test. When necessary, the post hoc analysis using Holm-corrected pairwise comparison was applied. The correlation between two continuous variables was calculated using Spearman’s correlation coefficient.

All analyses were performed in R for Windows, version 4.3 [26]. The *p* < 0.05 was selected as the threshold of significance.

## 4. Results

The analysed group of patients who died of COVID-19 comprised 477 patients aged 27–99 years. The group was predominantly male (283—59.3%). The full age structure is shown in Figure 1. Detailed data on the characteristics of the overall group are included in Table 1.

All patients required supplemental oxygen therapy, while 46 (9.6%) patients received baseline HFNOT or IMV. Twenty-one (4.4%) patients did not receive LMWH during the entire hospitalisation, five patients (1%) did not receive any anticoagulant treatment, while the remaining patients took oral medication (dabigatran, rivaroxaban, acenocumarol or warfarin). For those patients who received the therapeutic dose, the dosage change was mostly due to an increase in D-dimer levels (66 patients, 64.1%) or confirmed thromboembolic disease (24 patients, 23.3%). In the remaining patients (13, 12.6%), the reason for the dosage change was unknown.

### 4.1. Coinfections

More than half of the patients, 274 (57.4%), suffered from concomitant bacterial or fungal infections. In 159 (58%) of these patients (33.3% of all patients), HAIs were responsible (BSI, VAP, UTI). Fungal pathogens were responsible for 10.2% (28) of the coinfections. In blood cultures, the following were most commonly cultured: CNS—coagulase-negative *Staphylococcus*, including MRCNS, methicillin-resistant coagulase-negative *Staphylococcus* was 23.0% (38); *Klebsiella pneumoniae* ESBL (extended-spectrum beta-lactamases) 15.8% (26); *Acinetobacter baumanii* (most commonly, MBL, metallo beta-lactamases), 8.5% (14); and additionally, *Staphylococcus aureus*, including MRSA (methicillin-resistant *Staphylococcus aureus*), *Staphylococcus haemolyticus*, *Staphylococcus lugdunensis*, *Staphylococcus simulans* and *Staphylococcus capitis*, as well as enterococci, such as *Enterococcus faecalis*, including HLAR (high-level aminoglycoside resistance) and VRE (vancomycin-resistant Enterococcus), *Enterococcus faecium* and *Enterococcus cloacae*; *Pseudomonas aeruginosa*; *Propionibacterium acnes* and the fungi *Candida glabrata* and *Candida krusei*. 

In the cultures of bronchial secretions, the following were most commonly cultured: *A. baumanii* (including MER/IMI—meropenem/imipenem resistance and MBL) at 35.3% (61), *K. pneumoniae* ESBL at 10.4% (18), *P. aeruginosa* at 6.9% (12) and, additionally, *S. aureus* (including MRSA), *S. haemolyticus*, *S. simulans*, *Staphylococcus hominis* and CNS (including MRCNS), as well as *Streptococcus agalactiae*, *E. faecalis* (including HLAR and VRE), *E. faecium*, *Escherichia coli* ESBL, *Stenotrophomonas maltophilia*, *Serratia marcenses*, *Kocuria kristinae*, *Pantoea agglomerans*, and the fungi *C. glabrata*, *Candida albicans* and *Candida tropicalis*. 

In contrast, the following urine cultures were most commonly cultured: *A. baumanii* (including MER/IMI and MBL)—19.0% (23), *Candida* spp. (*C. albicans*, *C. glabrata*, *C. parapsilosis*)—12.4% (15), *E. faecium*—8.3% (10) and, in addition, *E. faecalis* (including HLAR and VRE), *Str. agalactiae*, *P. aeruginosa*, *E. coli* (including ESBL), *K. pneumoniae* (including ESBL and MBL). An analysis of the structure of coinfections and antibiotic use is shown in Table 2.

Coinfections were more common among patients who were initially admitted to the ICU compared to the IDU (*N* = 46; 82.1% vs. *N* = 228; 54.2%; *p* < 0.001). There was also a higher prevalence of nosocomial infections in that group (*N* = 32; 57.1% vs. *N* = 127; 30.2%; *p* < 0.001). HAI was observed statistically significantly (*p* < 0.001) more often in the youngest patients compared to the ≥81 years group (*N* = 15; 57.1% vs. *N* = 24; 15.9%)—Appendix A. The use of an antibiotic at the beginning of hospitalisation did not significantly affect the difference in the incidence of nosocomial infection (*p* = 0.7)—seven (28.0%) patients in the non-antibiotic group and 152 (33.6%) in the group treated with an antibiotic. The analysis of COVID-19 therapies revealed that the use of baricitinib was associated with a higher incidence of nosocomial infections (*p* < 0.001) (Table 3).

### 4.2. COVID-19 Complications

Table 4 provides data on COVID-19 complications, which are divided into three groups: thrombotic complications, bleeding complications and other complications.

### 4.3. Complications Unrelated to COVID-19

This group assessed events that were related to the patient’s hospital stay, including medical personnel malpractice (Table 5). A lack of adequate specialist treatment was highly likely to contribute to patient death, while a delay in specialist treatment was one of the additional factors likely to influence death.

Personnel malpractice included the failure or delay in intensifying treatment for respiratory failure, including delayed intubation, transfer of the patient to the ICU, failure to administer immunomodulatory drugs in case of signs of a “cytokine storm”, failure to administer antibiotic therapy in the face of a confirmed bacterial infection and inappropriate anticoagulant treatment.

### 4.4. Causes of Death

Respiratory failure due to SARS-CoV-2 infection was the predominant cause of death in 423 (88.7%) patients, with a significant advantage (*p* = 0.003) in the group admitted in the WHO’s stage three and four (318—91.4%) compared to stage one and two (105—81.4%). Out of 54 (11.3%) cases in which the predominant cause of death was not respiratory failure, 47 (87.0%) cases presented with sudden cardiac arrest (SCA) without preceding severe respiratory failure or signs of hypoxaemia. Figure 2 shows the analysis of other concomitant causes of death. The sum of the cases from Figure 2 does not equal 54 (11.3%) because some of the patients who died of COVID-19-associated respiratory failure also had additional relevant factors that influenced this and could not be omitted.

In terms of the other concomitant causes of death (Figure 2), cases in which the patient’s underlying condition (e.g., chronic renal failure on dialysis or malignant neoplasm on systemic therapy) could not be adequately treated due to SARS-CoV-2 infection were included in the “lack of specialist treatment” group. Cases in which ICU treatment was not provided due to the lack of places or low priority for such treatment were also collected under this section. Cases that could not be categorised included: upper gastrointestinal bleeding in a patient with decompensated cirrhosis and hepatocellular carcinoma, pulmonary oedema without an established clear cause, aspiration pneumonia, severe ischaemic and haemorrhagic strokes, subarachnoid haemorrhage, neuroCOVID with myoclonus syndrome and AIDS with pulmonary tuberculosis.

In a further analysis of the collected material, comparisons were made between the different age groups of patients without distinguishing between the sexes (in Appendix A) and virus variants in Table 6 and Figure 3. Therapies that were only used in the first pre-Delta pandemic phase (COVID-19 convalescent plasma and chloroquine + LPV/r) were not assessed in the analysis of virus variants.

## 5. Discussion

The analysis of 477 cases of SARS-CoV-2 infection resulting in death revealed that this group was dominated by elderly men with multimorbidity, who were admitted in a severe or very severe course of the disease according to the WHO classification, with low saturation values (<90%) and laboratory abnormalities indicating the presence of increased inflammation and organ damage. Patients aged ≥ 81 years were significantly more likely to be admitted to and die in infectious diseases units due to respiratory failure, their hospital stays being shorter, and they were most likely not to receive specialist treatment. More than half of the patients suffered from concomitant bacterial and fungal infections, and one-third had HAIs, and the initial use of an antibiotic, although common, was unwarranted in almost half of the cases. The inclusion of baricitinib significantly influenced the occurrence of HAIs, which was not found for the other COVID-19 therapies. The use of dexamethasone was significantly more frequent when the Delta variant was predominant. In terms of COVID-19 complications, AKI and thrombotic complications were the most common. In almost half of the cases, complications of hospitalisation not directly related to COVID-19 were found, and a similar proportion was of medical personnel malpractice. A cause of death other than respiratory failure affected more than 10% of patients.

The currently known risk factors for severe COVID-19 and death from COVID-19 show a clear association with an age > 65 years, male sex, comorbidities and the viral variant being Delta [6,8,9,27,28,29,30]. In the analysed group, the results were obtained and are in line with reports by other authors. More than half of the cases were patients aged >70 years. The presence of comorbidities was common (91.4%) and were mostly cardiovascular diseases (76.9%) and diabetes (33.5%). Given the age structure of the patients, such results are not surprising. The male predominance was significant (59.3%) but decreased with age (in the <50-year-old group, the male proportion was 76.2%, while in the ≥80-year-old group, it had already dropped to 51%), which may be influenced by the hormonal differences, ACE2 protein expression and a change in immune function with age. Similar relationships were described in publications on infection with other coronaviruses (SARS—severe acute respiratory syndrome; MERS—Middle East respiratory syndrome) [31,32,33]. 

The analyses of selected laboratory parameters indicate the presence of increased inflammation and vascular abnormalities already at the time of hospital admission—this is illustrated by the high mean CRP, PCT, ferritin and D-dimer levels, as well as creatinine and LDH levels, and the high mean and median EASIX that was identified as an independent prognostic factor for a severe course of COVID-19 and risk of COVID-19-related death [34,35,36]. This factor is derived from the calculation of LDH, creatinine and PLT, whose abnormal values reflect, among other things, vascular endothelial cell damage, which is one of the most significant processes leading to microangiopathy, thrombotic complications and organ failure in severe COVID-19 [37,38,39,40]. In the material evaluated, the mean (6.22) and median (3.14) EASIX values were above the cut-off points set in other publications analysing this factor in the light of its predictive ability of ICU hospitalisation and death, including one of our publications (Luft et al. EASIX ≥ 2.03, Kalicińska et al. EASIX ≥ 1.6 and Zińczuk et al. EASIX ≥ 2.36) [34,35,36].

Another significant finding of the analysis is the higher prevalence of coinfections and HAIs in younger age groups (*p* < 0.001). Coinfections were found twice as often in the <50-year-old group compared to the ≥81-year-old group (71.4% vs. 35.8%), and HAIs were more than three times as common (57.1% vs. 15.9%). It is possible that such a relationship is related to more frequent hospitalisations in the ICU, where microbiological tests are performed more frequently, and the risk of additional infections is higher than in the IDU [41]. The nearly two-fold shorter duration of hospital stays in the ≥81-year-old group, with a mean of 9.72 days (median 8 days (4–14), compared to the <50-year-old group, is also significant. A study by Contou et al. found that the median survival in critically ill patients with COVID-19 was 14 days after admission to the ICU (9–23) [42]. Our analysis provides data on the overall hospital length of stay of patients, with a median length of stay of 11 days (6–19) over the entire time interval analysed, which is a significantly worse result; however, it is difficult to identify a single reason for this.

A comparison of the course of the pandemic, according to the predominance of a particular viral variant in the Polish setting, is hampered by the lack of capacity to sequence SARS-CoV-2 genetic material in most healthcare institutions. Some publications on the evaluation of individual variants used the GISAID database, which provided reliable data on the dominance of specific viral sequences in a given region at a specific time [18]. In analogy with other publications from the Polish region, there was a division into periods of dominance of the pre-Delta, Delta and Omicron variants [8,9]. The analysis revealed a higher mean age of patients as the pandemic progressed (*p* = 0.015) when comparing the pre-Delta and Omicron variants (Table 6). 

The analysis of the infection data produced results confirming the high importance of this problem in COVID-19 patients. A coinfection was found in more than 50% of the patients (57.4%) who died, including one-third of patients (33.3%) who suffered from HAIs, and more often in those who were admitted to the ICU (Table 2). In studies by other authors, the prevalence of coinfections in the general population of COVID-19 patients was less than 7–14%, with a much higher proportion of ICU-acquired infections (even more than 50%) and a much higher ICU mortality rate among patients with coinfections [43,44,45,46]. Given that more than 55% of the deaths in our evaluation of the material were in the ICU, this finding may be comparable with those studies. However, the results obtained should be assessed with caution, as each time, the decision to perform cultures of biological material was assessed by an attending physician and thus was not part of the standard management of a SARS-CoV-2 infected patient in IDU; therefore, differences in terms of the number, timing and type of cultures performed are difficult to compare. Importantly, the vast majority of cultures were from patients in the most severe condition—those hospitalised in the ICU. This determines the expected higher frequency (up to five-fold for stays >1 week) of bacterial isolation, including drug-resistant pathogens [43]. The type of pathogens isolated is in agreement with data on the profile of isolated pathogens in ICUs [47,48,49].

The overuse of antibiotics proved to be a major problem in the treatment of COVID-19 patients. In the analysed group, up to 94.8% of patients were initially treated with antibiotics, while according to our own evaluation of the medical records, indications for the actual use of antibiotics only occurred in approximately half of the cases. A positive aspect of the discussed issue is that as the pandemic progressed and more studies discussing the issue became available, there was a significant (*p* < 0.001) decrease in antibiotic use, from the initial 97.6% for the pre-Delta period to 83.5% during the Omicron period (Table 6). The WHO guidelines (current as of 13 January 2023) indicate the need for an appropriate antibiotic policy among COVID-19 patients: refraining from using antibiotics in cases of mild disease and being guided by the clinical picture and laboratory parameters indicating the need for additional antimicrobial therapy among the remaining patients [25].

From the very beginning of the pandemic, there was a search for therapeutic approaches focused on the causal treatment of SARS-CoV-2 infection. The first therapies were based on reports concerning the mechanisms of action of previous epidemic coronaviruses causing infections among humans (SARS and MERS), identifying specific targets for drugs [50,51,52]. In the pre-Delta period, convalescent plasma, chloroquine plus lopinavir/ritonavir and remdesivir were used as part of the clinical trial. With the exception of remdesivir, the other therapies were discontinued in subsequent waves of the pandemic. Immunomodulatory treatments included dexamethasone, tocilizumab and baricitinib. Dexamethasone was the most widely used drug—almost 80% of the patients received it during their hospital stay, with a significant (*p* < 0.001) prevalence during the Delta variant-dominant period (97.3%) (Table 6). Numerous randomised trials, which were also evaluated in meta-analyses, produced results supporting the validity of glucocorticosteroid use among COVID-19 patients [53,54]. The frequency of use of other drugs among the cases evaluated in this study did not depend on the pandemic phase, except for baricitinib (*p* < 0.001). The therapies used against SARS-CoV-2 mostly did not affect the risk of HAIs, with the exception of baricitinib (Table 3), an inhibitor of the enzymatic activity of the JAK1 and JAK2 kinases, which conferred a significantly higher (*p* < 0.001) risk of HAIs. Contradictory to this result, however, is the data from the RECOVERY study and an additional meta-analysis of nine randomised trials on the use of JAK inhibitors in the treatment of COVID-19—no higher infection rate was found in the group of patients receiving these drugs compared to the standard therapy [55]. The differences may be due to the use of baricitinib in our setting in patients with severe respiratory failure who were no longer eligible for RDV therapy or who had completed it. 

All patients had supplemental oxygen therapy modalities; however, the oxygen therapy was not, in all cases, the least invasive oxygen therapy at baseline, involving the use of an oxygen mask with an oxygen flow of at least 10 L/min. In approximately 10% of the cases (9.6%), HFNOT was implemented, or endotracheal intubation and mechanical ventilation were performed immediately after the patient was admitted to the emergency room—this concerned the most severely ill patients with profound respiratory failure. This management, dictated by necessity, is also justified by the available literature—patients in whom standard oxygen therapy is ineffective should have HFNOT implemented, resulting in fewer ventilation complications and the need for intubation [25,56,57,58].

A severe SARS-CoV-2 infection is associated with cytokine abnormalities and over-stimulation of the inflammatory response, termed a cytokine storm [59]. Complications that are directly related to this viral infection are thus most often the result of vascular endothelial damage, which results in a higher risk of thromboembolic disorders and, at a later stage, organ damage [60]. In the evaluated group, such complications were confirmed in 112 patients, representing less than 25% of cases. Venous thromboembolism was the most common, found in 46 patients (9.6%). However, these figures may be underestimated due to the lack of technical feasibility of performing a CT in some patients in severe conditions with high oxygen demand. The literature data placed the prevalence of such disorders at 16.5% (95% CI 11.6–22.9) [61]. The most common COVID-19 complication that was observed in this study was AKI, found in 149 patients (31.2%). The meta-analysis by Yang et al. concludes that this result is comparable to global data. In a general population analysis of COVID-19 patients, AKI occurs at a frequency of 12.3% (95% CI 9.5–16.5%), with a much higher rate among cases hospitalised in the ICU being 39.0% (95% CI 23.2–57.6%) and those ending in death being 42.0% (95% CI 30.3–54.7%) [62].

The attempted analysis of adverse events yielded a surprising result—in almost half of the COVID-19 deaths assessed, adverse events in the course of hospitalisation could be identified. The unwarranted use of antibiotics, as discussed above, was one of those adverse events. The assessment of this type of event, in terms of a new aetiological agent, overload of the healthcare system and a lack of clear treatment guidelines, poses many difficulties. Based on the available data, it was not possible for us to establish the root cause of the events that occurred or, in many cases, assess the real impact on survival. An unambiguous analysis of the processes leading up to the incident would require a much larger study using incident reports [63]. Many therapeutic failures are certainly not described in the medical records. Our data, although indicative, show adverse events as an important risk factor for therapeutic failures, which is worth exploring in further studies. Medical personnel malpractice was found in a total of 49.7% of cases. Lack of specialist treatment (17.0%) in some patients may have contributed significantly to death. However, there was a more frequent (32.7% of cases) delay in the use of various therapeutic modalities, including endotracheal intubation and immunomodulatory treatment, which was influenced by the limited access to places of hospitalisation in the ICU and a lack of medication. The literature data on medical errors and adverse events in the era of the COVID-19 pandemic are limited. Prior to the onset of SARS-CoV-2-related morbidity, estimates from the USA in 2017 indicated that medical errors may be the third predominant cause of death in the country [64]. In high-income countries, adverse events occur, on average, in 1 in 10 patients [65]. Estimates for low- and middle-income countries indicate that up to one in four patients are harmed, and 134 million adverse events occur annually and are responsible for 2.6 million preventable deaths [66]. As many as 60% of deaths in these countries can be attributed to a failure to maintain safety in the delivery of health services and a poor quality of care [67].

Working under the strain of a new, hitherto unprecedented situation faced by the medical staff who were directly involved in the fight against the pandemic may have led to the occurrence of more errors, especially with the scarcity of adequate infrastructure and resources. A large number of severely ill patients requiring mechanical ventilation confronted the physician with the choice of which patient had a higher priority for such treatment [68]. During the COVID-19 pandemic, patient safety became a particular challenge. Numerous organisational factors, staff shortages, redeployment and temporary solutions disrupted the operation of most healthcare systems worldwide. The quality of services declined due to the delays in the implementation of treatment for both COVID-19 and all other conditions due to the lack of or limitations in the availability of services. There were also new types of diagnostic errors that were variously related and unrelated to the SARS-CoV-2 virus infection itself [69].

The WHO, in a document dated 7 June 2020, provided details on defining a COVID-19 death: it is considered to be “a death resulting from a clinically compatible illness, in a probably or confirmed COVID-19 case, unless there is a clear alternative cause of death that cannot be related to COVID disease” [70]. The evaluation of our patient group found that respiratory failure in the course of infection represented the vast majority of deaths (88.7%), while the remaining 11.3% deaths were of other causes representing conditions that resulted directly from infection (mainly SCA without preceding hypoxaemia) but also deaths in patients “with COVID-19 not due to COVID-19”. This group included, among others, patients who, due to a positive test for SARS-CoV-2 infection (without symptoms of respiratory failure), could not have continued specialist therapy in the absence of an adequately prepared infrastructure, e.g., dialysis, surgery or the systemic treatment of cancer. In the few available publications evaluating the causes of death, the results were comparable to ours. A retrospective single-centre study by Cobos-Siles et al. found that in 15% out of 128 patients, death was due to another cause with SARS-CoV-2 infection without developing ARDS or other complications directly associated with the viral infection. This group of patients was older (median 88 years) and had a higher multimorbid burden [71]. In contrast, Slater et al., in their report evaluating 162 deaths from one hospital in Leeds, indicated that SARS-CoV-2 infection was the direct cause of death in 93% of patients (“deaths due to COVID-19”); data were based on medical reports without defining a COVID-19 death to the end. It was indicated that 92% of patients had pulmonary infiltrates, and 97% required oxygen therapy [72]. Discrepancies in the correct classification of death during the pandemic may be due to the problem of over-estimation at the start of the pandemic and under-estimation of COVID-19 deaths as the pandemic progressed. It is emphasised that the COVID-19 pandemic is actually a syndemic, where the majority of deaths occur in patients with additional disease burdens, and sometimes determining which factor actually led to a patient’s death can be difficult, especially with the decline in the number of autopsies performed [73,74,75].

The analysis we performed has several limitations. Firstly, this analysis is retrospective and concerns only COVID-19 cases that ended in death, with no comparison to the group of patients who survived. Secondly, this is not a multicentre study, which means that local conditions, such as the standard of care, access to specific drugs and therapies, and the profile of patients admitted, may differ from the population average (as the standard of care is higher in a centre specialising in infectious diseases than in some hospitals). The analysis only used available medical records, which could also provide an erroneous assessment of the event in question, especially in the case of a delay or lack of specialist treatment, as well as in determining the final cause of death and the type of complications of hospitalisation.

## 6. Conclusions

COVID-19 is a disease that causes deaths mainly among elderly patients with multiple disease burdens. Respiratory failure due to SARS-CoV-2 infection is the most common mechanism of death, while sudden cardiac arrest and deaths due to thromboembolic complications occur less frequently. The hospitalisation of patients with severe COVID-19 is associated with numerous adverse events, which were only, in part, dependent on medical personnel and were largely related to the shortcomings of the healthcare system or resulted from a stay in a hospital setting. In light of possible future pandemics caused by similar viruses, it should be noted that mechanisms need to be developed for an efficient health system response to the rapidly increasing care needs of critically ill patients, including the elimination of preventable deaths, the implementation of appropriate management standards and an increase in patient safety.

## Figures and Tables

**Figure 1 viruses-15-01430-f001:**
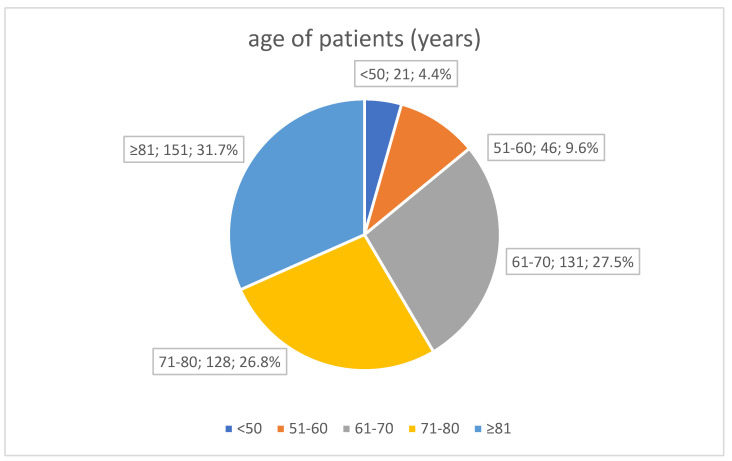
Age of patients (years).

**Figure 2 viruses-15-01430-f002:**
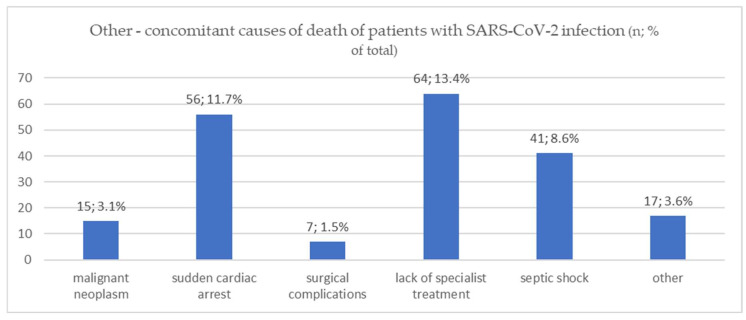
Other—concomitant causes of death of patients with SARS-CoV-2 infection (*N*; % of total).

**Figure 3 viruses-15-01430-f003:**
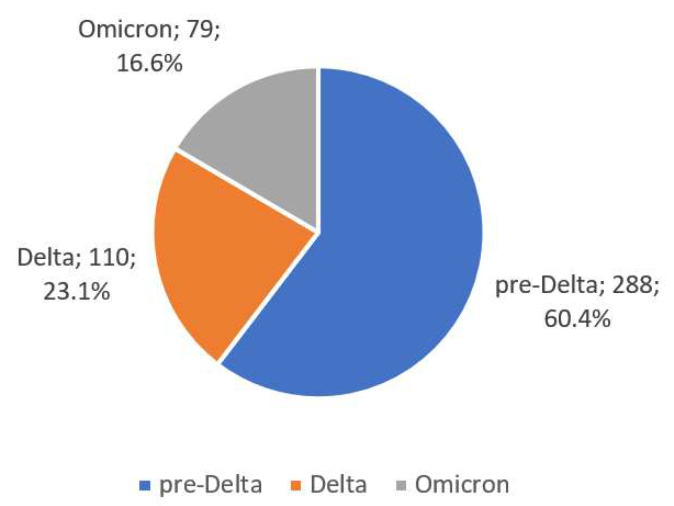
SARS-CoV-2 virus variants in the analysed group.

**Table 1 viruses-15-01430-t001:** Group characteristics. Quantitative parameters and variables are presented as mean values (standard deviation—SD) and median [interquartile range—IQR], while qualitative variables are presented as frequency/number. Normal laboratory parameters are given in brackets next to each value.

Variable	*N* = 477
Age (years)	73.43 (12.43), 74 [6–83]
Sex	female 194 (40.7%)
	male 283 (59.3%)
Ward of admission	ICU 56 (11.7%)
	IDU 421 (88.3%)
Ward where the patient died	ICU 264 (55.3%)
	IDU 213 (44.7%)
Length of hospitalisation (days)	13.87 (10.97), 11 [6–19]
Duration of symptoms before admission (days)	7.3 (4.62), 7 [5–9]
SpO_2_ at admission (%)	81.33 (13.5), 85 [74.75–91]
Severity of illness at admission acc. to WHO	1—14 (2.9%) 2—115 (24.1%) 3—236 (49.5%) 4—112 (23.5%)
Comorbidities: Cardiovascular diseases Respiratory diseases Diabetes Malignant neoplasm Chronic kidney disease Obesity	436 (91.4%) 367 (76.9%) 68 (14.3%) 160 (33.5%) 90 (18.9%) 58 (12.2%) 92 (19.3%)
Laboratory results at admission	
CRP (>6 mg/L)	128.2 (95.76), 105.44 [54.07–180.67]
D-dimer (>500 ng/mL)	4221.07 (9138), 1766 [1055–3278]
Ferritin (>274.66 ng/mL)	2279.94 (3887.55), 1292.69 [613.75–2690.47]
Creatinine (>1.15 mg/dL)	1.58 (1.63), 1.10 [0.82–1.67]
WBC (4–10 G/L)	9.58 (10.78), 7.64 [5.25–10.92]
PLT (150–420 G/L)	206 (100), 188 [136–248]
PCT (>0.05 ng/mL)	2.33 (10.18), 0.32 [0.12–0.97]
LDH (>220 U/L)	578.78 (304.16), 510 [368–703]
EASIX	6.22 (11.19), 3.14 [1.89–6.07]
Treatment during the entire hospitalisation	
Low-dose oxygen therapy > 10 L/min	431 (90.4%)
HFNOT	265 (55.6%)
NIV	132 (27.7%)
IMV	273 (57.2%)
Antiviral treatment (RDV, MPV)	94 (19.7%)
Chloroquine + LPV/r	29 (6.1%)
COVID-19 convalescent plasma	81 (17%)
Dexamethasone	381 (79.9%)
Tocilizumab	95 (19.9%)
Baricitinib	49 (10.3%)
Low-molecular-weight heparin	none—21 (4.4%) prophylactic dose—353 (74%), therapeutic dose—103 (21.6%)
Coinfections	274 (57.4%)
Healthcare-associated infections	159 (33.3%)
Predominant cause of death—COVID-19 respiratory failure	423 (88.7%)

**Table 2 viruses-15-01430-t002:** Characteristics of coinfections.

Variable	*N* = 477
Coinfection	274 (57.4%)
Healthcare-associated infection	159 (33.3%; 58% of all infections)
BSIs (bloodstream infections)	165 (34.6%)
VAP (ventilator-associated pneumonia)	173 (36.3%)
UTIs (urinary tract infections)	121 (25.4%)
Use of antibiotics from the beginning of hospitalisation	452 (94.8%)
Inclusion of an antibiotic without medical indication	215 (47.6%)

**Table 3 viruses-15-01430-t003:** The relationship between COVID-19 treatment administered and the occurrence of nosocomial infection.

COVID-19 Therapy	Hospital-Acquired Infection [*N* (%), *p*-Value]
Anti-viral treatment (RDV, MPV) (*N* = 94)	27 (28.7%), *p* = 0.3
Chloroquine + LPV/r (*N* = 29)	12 (41.4%), *p* = 0.4
Dexamethasone (*N* = 381)	117 (30.7%), *p* = 0.052
Tocilizumab (*N* = 95)	32 (33.7%), *p* > 0.9
Baricitinib (*N* = 49)	33 (67.3%), *p* < 0.001

**Table 4 viruses-15-01430-t004:** COVID-19 complications.

Variable	*N* = 477
Thrombotic complications:	112 (23.5%)
Venous thromboembolism	46 (9.6%)
Ischaemic stroke	29 (6.1%)
Myocardial infarction	22 (4.6%)
Limb ischaemia	16 (3.4%)
Bleeding complications	71 (14.9%)
Other:	261 (54.7%)
Acute kidney injury (AKI)	149 (31.2%)
Exacerbation of chronic heart disease	100 (21%)
Decompensation of liver function	12 (2.5%)

One patient with saddle pulmonary embolism also had an ischaemic stroke.

**Table 5 viruses-15-01430-t005:** Adverse events during hospitalisation of COVID-19 patients.

Variable	*N* = 477
Complications of hospitalisation	220 (46.1%), including more than one complication in 22 patients (4.6%)
Healthcare-associated infection	159 (33.3%)
Emphysema and other complications of ventilation	36 (7.5%)
Trauma, fall	25 (5.2%)
Personnel malpractice	237 (49.7%)
Lack of specialist treatment	81 (17.0%)
Delay in specialist treatment	156 (32.7%)

**Table 6 viruses-15-01430-t006:** The characteristics of selected parameters according to virus variant quantitative parameters and variables are presented as mean values (standard deviation—SD) and median [interquartile range—IQR], while qualitative variables are presented as frequency/number.

Variable	Pre-Delta (*N* = 288)	Delta (*N* = 110)	Omicron (*N* = 79)	*p*-Value
Age (years)	72.34 (11.9), 72 [65–82]	74.71 (13.23), 76 [66–85.75]	75.63 (12.86), 77 [69–85.5]	0.015
Sex				0.065
Men Women	183 (63.5%) 105 (36.5%)	57 (51.8%) 53 (48.2%)	43 (54.4%) 36 (45.6%)	
Duration of symptoms before admission (days)	6.94 (4.44), 7 [4–8]	7.44 (3.53), 7 [5–9]	8.83 (6.32), 7 [4.25–14]	0.093
Length of hospitalisations (days)	14.01 (11.17), 12 [6–20]	13.99 (9.90), 11.5 [7–19]	13.16 (11.17), 11 [5–16]	0.5
Comorbidities	266 (92.4%)	97 (88.2%)	73 (92.4%)	0.4
Cardiovascular diseases	219 (76.0%)	81 (73.6%)	67 (84.8%)	0.15
Respiratory diseases	42 (14.6%)	9 (8.2%)	17 (21.5%)	0.031
Diabetes	113 (39.2%)	29 (26.4%)	18 (22.8%)	0.004
Malignant neoplasm	57 (19.8%)	15 (13.6%)	18 (22.8%)	0.2
Chronic kidney disease	41 (14.2%)	9 (8.2%)	8 (10.1%)	0.2
Obesity	71(24.7%)	16 (14.5%)	5 (6.3%)	<0.001
EASIX	6.67 (12.63), 3.17 [1.91–6.23]	5.14 (5.28), 3.28 [2.05–6]	5.64 (9.97), 2.69 [1.66–4.45]	0.3
Treatment				
Antiviral treatment (RDV, MPV)	67 (23.3%)	16 (14.5%)	11(13.9%)	0.054
Dexamethasone	208 (72.2%)	107 (97.3%)	66 (83.5%)	<0.001
Tocilizumab	51 (17.7%)	29 (26.4%)	15 (19.0%)	0.2
Baricitinib	2 (0.7%)	30 (27.3%)	17 (21.5%)	<0.001
Antibiotic use	281 (97.6%)	105 (95.5%)	66 (83.5%)	<0.001
Inclusion of an antibiotic without medical indication	133 (47.0%)	58 (55.2%)	24 (36.4%)	0.051
Coinfection	166 (57.6%)	65 (59.1%)	43 (54.4%)	0.8
COVID-19 complications				
Pulmonary embolism	22 (7.6%)	14 (12.7%)	10 (12.7%)	0.2
Ischaemic stroke	17 (5.9%)	5 (4.5%)	6 (7.6%)	0.7
Myocardial infarction	13 (4.5%)	5 (4.5%)	4 (5.1%)	>0.9
Limb ischaemia	7 (2.4%)	2 (1.8%)	7 (8.9%)	0.028
Bleeding complications	39 (13.5%)	17 (15.5%)	15 (19.0%)	0.5
Acute kidney injury	86 (29.9%)	33 (30.0%)	30 (38.0%)	0.4
Exacerbation of chronic heart disease	60 (20.8%)	29 (26.4%)	11 (13.9%)	0.11
Decompensation of liver function	5 (1.7%)	1 (0.9%)	6 (7.6%)	0.011
Complications of hospitalisation				
Hospital-acquired infection	91 (31.6%)	42 (38.2%)	26 (32.9%)	0.5
Emphysema and other complications of ventilation	21 (7.3%)	12 (10.9%)	3 (3.8%)	0.2
Trauma, fall	8 (2.8%)	9 (8.2%)	8 (10.1%)	0.005
Predominant cause of death—COVID-19 respiratory failure	252 (87.5%)	103 (93.6%)	68 (86.1%)	0.15
Other causes of death				
Malignant neoplasm	10 (3.5%)	2 (1.8%)	3 (3.8%)	0.7
Sudden cardiac arrest	33 (11.5%)	10 (9.1%)	13 (16.5%)	0.3
Surgical complications	5 (1.7%)	2 (1.8%)	0 (0%)	0.7
Lack of specialist treatment	38 (13.2%)	14 (12.7%)	12 (15.2%)	0.9
Septic shock	14 (4.9%)	16 (14.5%)	11 (13.9%)	0.002
other	8 (2.8%)	3 (2.7%)	6 (7.6%)	0.14

## Data Availability

Data are available from the authors of the publication.

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
