# Peer review of "Unpacking the Complexity of COVID-19 Fatalities: Adverse Events as Contributing Factors—A Single-Center, Retrospective Analysis of the First Two Years of the Pandemic"

_viruses, 2023, doi:10.3390/v15071430_

Round 1
Reviewer 1 Report
The article is a concise retrospective analysis of fatal COVID cases at a single hospital from the beginning of the COVID-19 pandemic to the end of June 2022. The demographic of patients and array of comorbidities are well documented with all data tabled neatly and with appropriate statistical analyses.
The article is well written, and results are discussed thoroughly with limitations included in the discussion.
Authors could consider changing the title of the publication to include some more detail for readers; such as location, type of analysis and the duration of data collection.
Author Response
Dear Reviewer, thank you very much for your kind review and valuable comments.
Please find the answers to your comments below:
- Authors could consider changing the title of the publication to include some more detail for readers; such as location, type of analysis and the duration of data collection.
- The title of the publication has been modified as suggested.
Reviewer 2 Report
The Author reports in the manuscript entitled:"Unpacking the complexity of COVID-19 fatalities: adverse events as contribuiting factors" the results of a retrospective analysis of 477 fatal COVID-19 cases hospitalised at a single medical centre, during the period 2020 to 2022 with the aim to analyze the factors contributing to death.
The Author find that adverse events during hospitalisation is an additional factor that, may have influenced the death of patients.
My comments are:
1) The Author reports that the analysed group of patients who died of COVID-19 consisted of 477 patients, aged 27-99 years. Could be interesting know the comorbidities of young patients and the kind of acquired Healthcare-associated infections;
2) The Author describes that more than half of patients, (57.4%), suffered from concomitant bacterial or fungal infection: can the Author clarify the kind of fungal infections and the characteristics of patients , were they immunosoppressed?
3) The measure of EASIX is very interesting; did the Author find an improvement of the value of EASIX in patients treated with Dexamethasone?
Author Response
Dear Reviewer, thank you very much for the review and valuable comments.
Please find the answers to your questions below:
- The Author reports that the analysed group of patients who died of COVID-19 consisted of 477 patients, aged 27-99 years. Could be interesting know the comorbidities of young patients and the kind of acquired Healthcare-associated infections.
- The analysis of patient age groups was included in the supplementary materials (Table S1). The younger patients (<50 years) were grouped together due to the small size of sample, i.e. only 21 patients. The prevalence of comorbidities in this group was the lowest compared to the others (85.7%), but without a statistically significant difference (p=0.075). Patients in the age group <50 years (N=21) had the following conditions: 9 patients with obesity, 2 with chronic kidney disease including one with Goodpasture syndrome, 4 with malignancy (hepatocellular carcinoma, acute myeloid leukaemia, breast cancer), 4 with arterial hypertension, 4 with type two diabetes, 2 with asthma, 1 with alcoholic cirrhosis. - This information has been added under table S1 in the supplementary material.
- With regards to HAI, significant (p<0.001) differences in prevalence were found across age group, especially when comparing the oldest and the youngest age groups in a post-hoc analysis. The highest percentage of HAIs (57.1%) was found in the youngest patients. – information has been added in lines 199-200. These were mostly infections with A. baumanii MBL, K. pneumoniae ESBL, E.coli ESBL and isolated cases of S. aureus MRSA, E. facealis, Stenotrophomonas maltophilia.
- The Author describes that more than half of patients, (57.4%), suffered from concomitant bacterial or fungal infection: can the Author clarify the kind of fungal infections and the characteristics of patients, were they immunosoppressed?
- Fungal infections were found in 28 patients, representing 5.9% of patients in the analysed group (line 168). Microbiological examinations revealed only the Candida spp infections (lines 179, 186, 189). In our study, the immunosuppression status of patients was not assessed, due to the lack of a clear possibility to classify patients into this group, which is due to the many reasons affecting immunosuppression (e.g. oncology patients, patients with cirrhosis, using drugs with immunosuppressive effect). However, the majority of these were patients hospitalised in the ICU.
- The measure of EASIX is very interesting; did the Author find an improvement of the value of EASIX in patients treated with Dexamethasone?
- EASIX is a promising predictor of unfavourable course of COVID-19. In bulk of available literature, EASIX analysis is based on baseline laboratory parameters. Also in our study, the values of the parameters included in EASIX were not assessed in the treatment phase so the improvement in EASIX values was not assessed either. However, this is a promising direction for possible further research - assessing the effect of COVID-19 treatment agents on vascular endothelial dysfunction parameters expressed by EASIX.